# Salen-like Chromium and Aluminum Complexes as Catalysts in the Copolymerization of Epoxides with Cyclic Anhydrides for the Synthesis of Polyesters

**DOI:** 10.3390/ijms241210052

**Published:** 2023-06-13

**Authors:** Federica Santulli, Ilaria Grimaldi, Daniela Pappalardo, Marina Lamberti, Mina Mazzeo

**Affiliations:** 1Department of Chemistry and Biology “A. Zambelli”, University of Salerno, Via Giovanni Paolo II, 132, 84084 Fisciano, Italy; fsantulli@unisa.it (F.S.); ilagrimaldi@unisa.it (I.G.); mlamberti@unisa.it (M.L.); 2Dipartimento di Scienze e Tecnologie, Università del Sannio, Via de Sanctis snc, 82100 Benevento, Italy

**Keywords:** aluminum catalysts, block copolymers, chromium(III) catalysts, polyesters, ring opening copolymerization, switch catalysis

## Abstract

Chromium and aluminum complexes bearing salalen ligands were explored as catalysts for the ring-opening copolymerization (ROCOP) of succinic (SA), maleic (MA), and phthalic (PA) anhydrides with several epoxides: cyclohexene oxide (CHO), propylene oxide (PO), and limonene oxide (LO). Their behavior was compared with that of traditional salen chromium complexes. A completely alternating enchainment of monomers to provide pure polyesters was achieved with all the catalysts when used in combination with 4-(dimethylamino)pyridine (DMAP) as the cocatalyst. Poly(propylene maleate-*block*-polyglycolide), a diblock polyester with a precise composition, was obtained by switch catalysis, in which the same catalyst was able to combine the ROCOP of propylene oxide and maleic anhydride with the ring-opening polymerization (ROP) of glycolide (GA) through a one-pot procedure, starting from an initial mixture of the three different monomers.

## 1. Introduction

Aliphatic polyesters are an important class of degradable synthetic polymer, which currently represent the sustainable alternatives to petroleum-based polymers and are already used in different fields, from packaging to biomedical applications [1]. The large interest in the study of the synthesis and properties of this class of polymeric materials stems from different aspects, such as their potential origin from renewable sources, the hydrolytic degradation to benign products, their resorbable character, and their biocompatibility in certain cases [2].

While these features can also be found in some natural polymers, synthetic polymers are often favored over natural polymers. The synthetic approach, indeed, can offer the opportunity to tailor the microstructure, molecular weight, composition, end groups, and architecture of macromolecular chains. This aspect is mandatary when the material is intended for a biomedical application, such as tissue engineering, for the possibility of modulating the chemical and mechanical properties and degradation kinetics to suit various applications [3]. For all these reasons, aliphatic polyesters, such as polylactide, polyglycolide, poly(ε-caprolactone), and polyhydroxyalkanoates, are among the most frequently employed synthetic polymers for tissue-engineering applications [2,4].

The alternating ring-opening copolymerization (ROCOP) of epoxides and cyclic anhydrides is emerging as an effective route for the synthesis of polyesters with a large array of chemical and physical properties. The presence of two distinct monomers into the same repeating unit of the polymer chain allows the synthesis of materials with properties and functionalities that are not accessible through the strict ring-opening polymerization (ROP) of lactones [5,6,7]. Moreover, the combination of both these methodologies in switchable catalysis offers new opportunities for the efficient preparation of multiblock polyesters with original compositions by a single synthetic event, starting from monomer mixtures [8,9,10].

In the last decade, various metal complexes [11,12,13,14,15] and organic molecules [16] have been explored as catalysts in ROCOP for a variety of epoxides and anhydrides. In some cases, regio- and/or stereoselective copolymerization was achieved, producing stereoregular polyesters with enhanced physical properties [17,18].

This technique has also enriched the portfolio of biocompatible polyesters for biomedical applications with the inclusion of poly(propylene fumarate) (PPF), a linear polyester that satisfies a variety of important requirements for biomedical applications, including biocompatibility, osteoconductivity, and sterilizability. Furthermore, PPF, which is a linear polyester, non-toxic, and completely resorbable upon degradation, has recently attracted interest for biomedical uses due to its excellent biocompatibility and biodegradability [19]. Notably, upon degradation, fumaric acid, a constituent of the Krebs cycle, and propylene glycol are produced, which can be naturally excreted by the human body [20]. In addition, PPF is 3D-printable; the presence of one unsaturated carbon–carbon double bond in the repeating units can be exploited to form a UV-cross-linkable network, to strengthen the mechanical properties of the material, or to add functionalization to the macromolecular chains, by adding selected signals or clues to the surfaces of the materials [19,21].

Due to these interesting properties, the use of PPF as a degradable scaffold material for tissue engineering and drug delivery has been investigated since its introduction in 1994 [4,22].

The traditional synthetic method for PPF is the step-growth polycondensation of diethyl fumarate and propylene glycol; however, this approach is generally poorly controlled [19]. An alternative approach is the ring-opening copolymerization (ROCOP) of maleic anhydride (MA) and propylene oxide (PO), which can produce poly(propylene maleate) (PPM), in which the configuration of the double bond is *cis*; from the PPM, the *trans*-alkene isomer PPF can be easily obtained using a weak base at low temperatures.

Despite the great potential of this approach, currently, very few efficient catalysts are available. Coates described a cobalt catalyst able to produce well-controlled PPM that could be converted to PPF upon isomerization [23]. Subsequently, to prevent the toxicity of cobalt, magnesium ethoxide was described as a ROCOP catalyst to produce PPF, albeit with poorer control. Well-controlled ROCOP to give PPF was achieved in the presence of magnesium alkoxides as catalysts, resulting in molecular-weight distributions (*Đ*) similar to those reported by Coates et al. [21,24]. The same catalytic systems were used to produce various architectures of PPF copolymers, as well as terpolymers with different cyclic esters [25,26].

In the literature, salen-based chromium complexes, in combination with opportune cocatalysts, are among the most frequently used catalysts in the ring-opening co-polymerization of epoxides with anhydrides [27,28,29,30,31]. Although a substantial influence of the ligand structure on the catalytic behavior emerged in the studies of several authors, the effect of the higher coordinative flexibility of salan and salalen systems compared to rigid salen-based complexes or of the differences in donor ability between amine nitrogen and the related imine nitrogen have rarely been investigated [32].

In the framework of our interest in the copolymerization of epoxide with anhydrides [12,13,33,34,35,36], we recently compared salen, salalen, and salan chromium complexes bearing t-Bu substituents on phenolate moieties in the copolymerization of cyclohexene oxide and limonene oxide with phthalic anhydride [37].

In this work, we extended these studies by exploring the reactivity of pentacoordinate chromium and aluminum catalysts bearing salalen-type bisphenoxy ligands in the ROCOP of the most commonly used epoxides and anhydrides. Specifically, cyclohexene oxide (CHO) or propylene oxide (PO), which are traditionally obtained by the petrochemical route, were used and limonene oxide (LO) that is obtained through citrus-peel oil.

The anhydrides used were phthalic anhydride (PA) and succinic anhydride (SA), which are benchmark monomers for this type of polymerization. Subsequently, the copolymerization of MA with PO to produce PPM and the terpolymerization of MA with PO and glycolide to produce a block copolymer were investigated. The catalysts were used active in both the co- and the terpolymerization, opening the way to the potential use of novel polymeric materials in biomedical applications. 

## 2. Results and Discussion

### 2.1. Synthesis of Complexes

A systematic investigation of the copolymerization reactions between epoxides and anhydrides with salen complexes of trivalent metals (Al(III), Co(III), and Cr(III)) was reported by Duchateau et al. [30] In this study, the authors demonstrated that the catalytic activity of catalysts is strongly influenced not only by the nature of the metal center but also by the structure of the ancillary ligand. The effects of the presence of different substituents on the aryl rings and of the structure of the bridge between the two imine-nitrogen atoms were investigated, demonstrating that the backbone structure plays a predominant role.

In the light of these results, the first objective of this work was to verify the effect of the substitution of an imine-type neutral donor with an amino nitrogen atom on the activity and selectivity of chromium complexes.

The structures of chromium complexes (**Cr1″**–**Cr4**) and of the aluminium complex (**Al2**) investigated in this work are described in Figure 1.

The salen chromium complexes **Cr3** and **Cr4** were previously reported by Duchateau [30]. 

In this study, the chromium complex **Cr1** bore instead a salalen ligand, with the imine nitrogen substituted by an amine nitrogen; the **Cr1** was synthesized to compare its activity with analogous salen complexes, **Cr3** and **Cr4**, to evaluate the effect on the reactivity and selectivity of the copolymerization reaction of cyclohexene oxide with succinic anhydride. Specifically, in the chromium complex, **Cr1**, the ancillary salalen ligand had a structure analogous to that of the corresponding salen chromium complex, **Cr3**, i.e., tert-butyl groups in the ortho and para positions of the phenoxy rings [30]. 

Subsequently, we selected a salalen ligand with an amino nitrogen within a five-term cycle, which offered structural rigidity to the bridge between the two nitrogen atoms, and, at the same time, possessed a stereogenic center with an S configuration and was, therefore, chiral. The chromium **Cr2** and aluminum **Al2** complexes were synthetized with this chiral proligand to explore the possibility of stereoselective polymerization. 

The chromium complexes **Cr1**–**Cr3** were synthesized according to a previously reported procedure through the direct reaction of corresponding ligand precursors with CrCl_2_ in a THF solution (Figure 2) [37]. The reaction mixtures were stirred for 24 h at room temperature and exposed to air for an additional 24 h, to allow the oxidation of Cr(II) to Cr(III). The desired complexes were obtained after extraction with water and ethyl ether and were characterized by ESI spectrometry.

For complexes **Cr1** [37] and **Cr3** [38], the complete characterizations were reported in a previous publication [37].

In the ESI spectrum of the **Cr2** complex (Appendix A), the most intense peak (*m*/*z* = 581) was related to the ion (**Cr2** + ACN)^+^, i.e., the complex which lost a chloride ligand and gained an acetonitrile unit, the solvent used for the analysis. The signal at *m*/*z* = 558 corresponded to the (**Cr2** + H_2_O)^+^ ion, and the signal at *m*/*z* = 540 corresponded to the (Cr2)^+^ ion. All the observed peaks were indicative of the formation of the desired complex. 

The **Cr2** complex was analyzed by UV-visible spectroscopy, and its spectrum was compared to that of the free ligand. The samples were dissolved in acetonitrile (80 μmol) and the absorbance was recorded from 200 nm to 800 nm (Appendix A. Indicative of the formation of the desired chromium complex, a phenolate ligand-to-metal charge-transfer (LMCT) transition was observed: this band appeared at 426 nm, at lower energy and with lower intensity than the band in the spectrum of the corresponding free ligand. This was attributable to the π → π* transitions of the phenolic chromophores.

The IR spectrum of the ligand and the **Cr2** complex is reported Appendix A. In the spectrum appeared two peaks around 550 cm^−1^ and 490 cm^−1^, indicative of the M–N and M–O bonds, respectively.

The **Cr4** complex is a commercial product, previously investigated as a catalyst by Darensbourg for the alternating copolymerization of cyclic anhydrides and epoxides, and for terpolymerization in the presence of additional CO_2_ [31].

The aluminum complex **Al2** was previously reported by our group for the synthesis of polylactide featuring an original gradient isotactic multiblock microstructure, resulting from a combination of enantiomorphic-site and chain-end control mechanisms [39]. It was obtained, in an almost quantitative yield, through a direct reaction between the proligands with one equivalent of AlMe_3_ ( Figure 3) and was characterized by ^1^H and ^13^C NMR spectroscopy.

### 2.2. Polymerization of Epoxides and Anhydrides

All the complexes depicted in Figure 1 were tested as catalysts in the copolymerization reactions of several anhydrides and epoxides. The monomers investigated in this work are shown in Figure 4.

Most of the studies reported in the literature concern cyclohexene oxide (CHO) or propylene oxide (PO). Both of these monomers are traditionally obtained through the oxidation of petrochemical products. Recent studies have shown that CHO can also be obtained from renewable sources, starting from 1,4 cyclohexadiene, which is obtained through the metathesis of vegetable oils [40].

Limonene oxide (LO) is obtained through the natural cyclic monoterpene limonene as (R)-enantiomer, and it is commercially available as a mixture of the *trans* and *cis* diastereomers. Produced by more than 300 plants, it makes up 90–96% of citrus-peel oil, and its worldwide production is estimated to be between GBP 110 and GBP 165 million a year [41].

The anhydrides used were phthalic anhydride (PA) and succinic anhydride (SA), both of which are benchmark monomers for this type of polymerization. Succinic anhydride can be produced from renewable sources [42]. Furthermore, some studies have been reported concerning the obtainment of phthalic anhydride from bio-renewable sources [43].

Initially, the salalen chromium complexes **Cr1** and **Cr2** were tested in the copolymerization of succinic anhydride (SA) with cyclohexene oxide (CHO) under reaction conditions similar to those described in the literature for the salen chromium complex **Cr3** [30] and for salen aluminum complexes [11].

In these systems, a coordination-insertion mechanism is assumed to be operative, in which the polymerization is initiated by ring-opening of epoxide coordinated with the metal center by an external nucleophile (or by the labile ligand coordinate at the metal center) to provide a metal–alkoxide intermediate that undergoes an insertion reaction with the anhydride to afford a metal-carboxylate intermediate. The successive alternating incorporation of epoxide and anhydride produces linear polyester. In this mechanism, the rate-limiting step is the ring-opening of the epoxide by the enchained carboxylate end chain (Figure 5).

The polymerization was initially performed in the presence of two equivalents of 4-dimethylamino)pyridine (DMAP) as a co-catalyst at 110 °C. Both complexes **Cr1** and **Cr2** proved to be very active in the polymerization of CHO/SA, showing a turnover frequency (TOF) of 120 h^−1^ (entries 1 and 3, Table 1).

**Table 1 ijms-24-10052-t001:** Copolymerization of epoxides with anhydrides.

Entry	Cat	Ep.	Anhy.	T (°C)	Time (h)	Conv. ^[e]^ (%)	TOF (h^−1^)	Ester ^[e]^ (%)	M_n_ ^[f]^ (Kg/mol)	*Đ* ^[f]^
1 ^[a]^	**Cr1**	CHO	SA	110	2	95	119	88	1.3	1.31
2 ^[b]^	**Cr1**	CHO	SA	110	2	28	35	>99	1.1	1.42
3 ^[a]^	**Cr2**	CHO	SA	110	2	98	122	88	1.5	1.32
4 ^[c]^	**Cr3**	CHO	SA	130	2.5	100	100	-	1.9	1.25
5 ^[d]^	**Cr2**	CHO	SA	110	1	93	232	>99	2.3	1.18
6 ^[d]^	**Al2**	CHO	SA	110	2	81	101	>99	2.1	1.19
7 ^[d]^	**Cr2**	CHO	PA	50	24	100	10	>99	13.1	1.21
8 ^[d]^	**Al2**	CHO	PA	50	24	78	8	>99	9.9	1.11
9 ^[d]^	**Cr2**	PO	PA	50	24	60	6	>99	2.3	1.22
10 ^[d]^	**Al2**	PO	PA	50	72	73	2	>99	2.4	1.15
11 ^[d]^	**Cr2**	LO	PA	100	17	91	13	>99	3.6	1.21
12 ^[d]^	**Al2**	LO	PA	100	17	81	12	>99	3.0	1.10

^[a]^ General conditions: All reactions were performed by using 10 μmol of metal complex and the following ratios: epoxide:anhydride:metal:DMAP = 250:250:1:2 and 1 mL of toluene. ^[b]^ PPNCl was used as catalyst. ^[c]^ for details, see [30]. ^[d]^ Anhydride:epoxide:metal:DMAP = 250:1000:1:2. ^[e]^ Conv. (%) is the conversion of anhydride, and ester (%) is the percentage of the ester linkage in the polymer. Conversion of anhydride was determined by ^1^H NMR. ^[f]^ Experimental M_n_ and *Ð* values were determined by GPC analysis in THF using polystyrene standards.

This value was slightly higher than that obtained with the analogous salen-type complex, **Cr3** (TOF = 100 h^−1^, entry 4, Table 1), at 130 °C [30]. This difference is significant if we consider that in our case, the polymerization tests were conducted at lower temperatures.

Neither copolymerization was completely selective; ether linkages (12%), due to the subsequent insertions of epoxide units, were present in the polymer chains obtained with both catalysts. These were evaluated by the ^1^H NMR spectra, in which the resonances relative to the ether linkages were clearly evident in the region between 3.32 pm and 3.45 pm (see Figure 1).

Subsequently, a polymerization test with complex **Cr1** and bis(triphenylphosphine)iminium chloride (PPNCl) as the cocatalyst was performed (entry 2, Table 1). In this case, complete selectivity was observed, although the activity was significantly compromised.

To improve the control degree and to preserve the good catalytic activity, a new experiment was performed with DMAP, increasing the amount of CHO to 1000 equivalents (entry 5, Table 1), and in the absence of solvent. In this case, the ^1^H NMR spectrum of the polymer did not show consecutive anhydride or epoxide sequences, which suggests an alternated microstructure.

Subsequently, the chromium catalyst **Cr2** was selected for further investigations and compared with aluminium complex **Al2**. Both catalysts were tested in the ROCOP of succinic anhydride (SA) and phthalic anhydride (PA) with cyclohexene oxide (CHO), propylene oxide (PO), and limonene oxide (LO) (entries 5–12, Table 1).

Considering the chiral nature of the **Al2** and **Cr2** complexes, a microstructural analysis of the obtained polymers was carried out to verify the possibility of stereochemical control.

The CHO monomer is a *meso* isomer with two stereogenic centers of opposite chirality; a chiral catalyst can operate enantioselective discrimination between the two centers, producing a stereoregular polymer. Since the reaction proceeds with an inversion of the configuration of the stereogenic center on which the nucleophilic attack occurs with the SN2 mechanism, the polymer obtained is homochiral. Results of this type were recently reported by Lu et al., who described chiral bimetallic complexes of aluminium and chromium with salen ligands active in the regio- and stereo-regular copolymerization of CHO/PA [17,44].

For the samples described in entries 5–8 in Table 1, in the carbonyl region of the ^13^C NMR spectra, the presence of multiple peaks (about 167 ppm) was indicative of an atactic structure (see, for example, Appendix A).

The reaction rates in the copolymerization of the phthalic anhydride with different epoxides, observed under similar reaction conditions, provided relative reactivities of CHO > PO >> LO. No conversion of the monomers was detected for the LO/PA copolymerization performed at 50 °C, while discrete reactivity was observed at 110 °C.

In all cases, the obtained polymers showed alternating structures, with percentages of ester sequences in the chains higher than 99%, thus confirming the ability of both catalytic systems, **Cr2** and **Al2,** to operate selective polymerization under the selected reaction conditions. All the obtained polymers showed molecular masses lower than the theoretically expected values, although the distributions were always monomodal and characterized by rather narrow dispersities. This phenomenon is often observed in the copolymerization of epoxides with anhydrides, and it is attributable to the presence of protic impurities in the monomers, which act as chain-transfer agents.

In all the polymerizations investigated, the chromium **Cr2** catalyst showed higher activity levels than the analogous aluminium complex, **Al2**, while neither complex showed stereoselectivity.

### 2.3. Copolymerization of Maleic Anhydride and Propylene Oxide

The ROCOP of maleic anhydride (MA) and propylene oxide (PO) for synthesizing oligomeric poly(propylene maleate) (PPM), after conversion to poly(propylene fumarate) (PPF), provides a rapid and scalable method for synthesizing PPF with well-defined molecular masses, dispersity, and viscosity properties suitable for 3D printing. 

To achieve this, we tested the chiral chromium complex, **Cr2**, as catalyst in the copolymerization of maleic anhydride with racemic propylene oxide (Figure 6) and compared its behavior with that of the commercial complex, **Cr4** [23]. The polymerization reactions were initially performed at 45 °C and in hexane, as the solvent (entries 1 and 3, Table 2). The yields obtained were lower than those reported in the literature, as were the molecular masses.

Similar results were achieved when the copolymerization was performed in toluene solution and in the presence of an equivalent of PPNCl as the cocatalyst (entries 1 and 2, Table 2).

In all cases, alternating structures were obtained, as evidenced by the absence of the resonances characteristic of polyether sequences at 3.5 ppm in the ^1^H NMR spectra (Figure 2). The composition of the obtained polymer was dictated by its chemical structure; the product was a perfect propylene maleate due to the alternating enchainment of PO and MA. No ether sequences were observed, while the percentage of ester bonds was higher than 99%.

The molecular masses of the obtained polymers were very low; however, these values are optimal for 3D printing through stereolithographic processes. For this application, viscosities of less than 2 Pa s are needed to ensure the printability of structures.

Matrix-assisted laser-desorption-ionization mass spectroscopy (MALDI-ToF-MS) was used to precisely determine the masses of the individual polymers and the end-group populations. The MALDI-ToF-MS spectra of the obtained polymers showed two distinct series of peaks. Each series exhibited a *m*/*z* interval of 156 between consecutive peaks, which equals the masses of maleic anhydride and propylene oxide in the [PO + MA] repeating unit. The two distributions consisted of [PO + MA]_n_ and ([PO + MA]_n_ + PO) fragments, respectively (Figure 3).

Both the series were hydroxyl-functionalized. The chain-end groups were probably generated in the initiation step through the nucleophilic attack of the residual moisture traces present into the monomers, and in the termination step through the hydrolysis of the metal-chain bond of the propagating species.

This observation evidences the high stability of chromium catalysts against protic impurities and, at the same time, offers new opportunities for the synthesis of telechelic macromonomers.

The ring opening of PO may occur in methine or methylene carbon, giving rise to a series of possible regiosequences, as shown in Figure 4. The HT and TH junctions describe regioregular sequences, and they are indistinguishable by NMR spectroscopy. By contrast, the chemical shifts of the resonances corresponding the regioirregular HH and TT junctions were easily identified.

The regioregularity of the resultant PPMs was evaluated by the content of the head-to-tail (H–T) diads of PPM in the ^1^H and ^13^C NMR spectra [45] (Figure 4).

From the analysis of the spectra of the PPMs obtained, it emerged that neither the **Cr2** and nor the **Cr4** catalyst were regioselective; consequently, atactic polymers were obtained in all cases, as evidenced by the signals present at 130 ppm in the ^13^C NMR spectrum (Appendix A). 

Subsequently, the *cis*–*trans* isomerization of the MA-PO units was performed by using diethylamine (NHEt_2_) to convert the polypropylene maleate (PPM) to polypropylene fumarate (PPF) (Figure 7). A solution of the polymer in CDCl_3_ was stirred at room temperature in the presence of NHEt_2_, and the progress of the isomerization was checked by ^1^H NMR spectroscopy.

After 24 h, the ^1^H NMR spectrum (Figure 5) of the reaction mixture demonstrated the complete isomerization of the polymer from polypropylene maleate to polypropylene fumarate: the signal relative to the protons in the *cis*–alkene portions of the PPM was no longer present, and a new downshifted signal appeared for the methylene protons of the *trans*–alkene fragments of the PPF.

Finally, the performance of the chromium catalyst **Cr4** in the synthesis of block polyesters through the switch catalysis between the ROCOP of the maleic anhydride and the propylene oxide and the ROP of the glycolide was tested (Figure 8). Block and random copolymers of glycolide [46] are of great interest for biomedical applications, such as sutures, prosthetic devices, and drug-delivery systems.

The terpolymerization was carried out by using 2.0 mmol of maleic anhydride (MA), 2.0 mmol of glycolide (GL), and an excess of propylene oxide (1 mL). The mixture was then stirred at 50 °C. After 22 h, the ^1^H NMR spectrum showed the full conversion of both the maleic anhydride and the glycolide.

The composition of the copolymer was examined by ^1^H NMR spectroscopy (Figure 6), which exclusively revealed signals relative to the two blocks of polyesters, without the presence of additional resonances. Similarly, identical lengths of the two blocks were estimated. In the DOSY NMR spectrum (Figure 7), the presence of a single diffusion coefficient for the signals of the polyester fragments was consistent with the formation of a diblock polymer.

This hypothesis was also supported by the GPC analysis, which showed a single distribution of molecular masses with a *M*_n_ value of 3.5 KDa and a narrow dispersity *Ð* = 1.33 (Appendix A). This *M*_n_ value was lower than expected, as previously observed in other examples of switch catalysis ROCOP/ROP [11,13,33,34,35].

This evidence demonstrated that the chromium **Cr4** catalyst was able to switch between two different polymerization mechanisms and selectively enchain mixtures of commercially available monomers for the synthesis of a new diblock polyester with potential applications in biomedical fields.

To the best of our knowledge, this is the first example of the synthesis of a poly(propylene maleate)-*block*-polyglycolide using switch catalysis [47].

## 3. Materials and Methods

All the operations of synthesis and handling of air-sensitive chemicals were performed in an inert atmosphere, using Schlenk techniques and/or a glove-box in nitrogen atmosphere. The glassware was dried in an oven at 120 °C and subsequently subjected to vacuum–nitrogen cycles. Deuterated solvents were purchased from Sigma-Aldrich-Merck and dried over activated 3-Å molecular sieves prior to use.

Monomers (Sigma-Aldrich-Merck, Burlington, MA, USA) were purified prior to use: LO, CHO, and PO were distilled under vacuum on CaH_2_ and stored over 4 Å molecular sieves; phthalic anhydride (PA) was crystallized from dry toluene; glycolide was recrystallized from tetrahydrofuran. Toluene was acquired from Sigma-Aldrich-Merck; it was refluxed over Na and distilled under nitrogen. The CDCl_3_ was purchased from Eurisotop and used as received. The bis(triphenylphosphine)iminium chloride salt (PPNCl) and all other reagents and solvents were purchased from Sigma-Aldrich-Merck and used as received, unless stated otherwise.

The NMR spectra were recorded on a Bruker AM300 (^1^H, 300 MHz; ^13^C, 75 MHz) and a Bruker Avance 400 (^1^H, 300 MHz; ^13^C, 75 MHz). The resonances were reported in ppm (δ) and the coupling constants in Hz (J), and were referenced to the residual solvent peaks at δ = 7.27 (^1^H) and at δ = 77.23 (^13^C) for CDCl_3_. The NMR samples were prepared by dissolving about 10 mg of compound in 0.5 mL of the deuterated solvent. Spectra recording was performed using Bruker TopSpin v2.1 software. Data processing was performed using TopSpin v2.1 or MestReNova v6.0.2 software.

Molecular weights (Mn and Mw) and dispersities (Ð) of polymeric samples were measured by size-exclusion chromatography. The measurements were performed at 35 °C on an Agilent 1260 Infinity II HTGPC (RI) system equipped with a system of two PLGEL-column 5 µm MIXED-C (7.5 × 300 mm). Tetrahydrofuran was used as eluent at a flow rate of 1.0 mL min^−1^. Narrow polystyrene standards were used as references, and data processing was performed using Agilent GPC/SEC software A.02.02 (Agilent Technologies, Palo Alto, CA, USA).

The MALDI mass spectra were recorded using a Bruker solariX XR Fourier-transform ion-cyclotron resonance (FT-ICR) mass spectrometer (Bruker Daltonik GmbH, Bremen, Germany) equipped with a 7 T refrigerated and actively shielded superconducting magnet (Bruker Biospin, Wissembourg, France). The samples were prepared at the concentration of 1.0 mg mL^−1^ in THF, while the matrix (DCTB) was mixed at a concentration of 10.0 mg mL^−1^.

### 3.1. Synthesis of Complex Cr2

Salalen CrCl complex **Cr2** was prepared according to a published procedure [38]. To a 100 mL flask equipped with a magnetic stirrer, 332 mg (6.7 × 10^−4^ mol) of salen ligand and 91 mg (7.4 × 10^−4^ mol) of chromium(II) chloride CrCl_2_ were added. The two solids were dissolved in 50 mL of THF and the solution was stirred at room temperature in a nitrogen atmosphere for 24 h. The solution turned dark red/brown. The following day, the solution was exposed to the air for two hours to allow the oxidation of chromium(II) to chromium(III) and stirred for a further 24 h. In total, 50 mL of diethyl ether was added to the flask, and the solution was extracted with a saturated solution of NH_4_Cl (3 × 100 mL) and a saturated solution of NaCl (3 × 100 mL). The organic phase was dried with Na_2_SO_4_, filtered, and concentrated. The product was a brown-green powder. Yield: 80%. Characterization of **Cr2** was performed by MS (MALDI-ToF) *m*/*z*: 558,325 (Cr[Salen]^+^), 574,320 (Cr[Salen] + H_2_O), 590,315 (Cr[Salen] + O_2_).

### 3.2. Synthesis of Complex Al2

In a MBraun glovebox, to a stirred solution of AlMe_3_ (0.029 g; 0.4 mmol) in benzene (1 mL), a solution of the proligand (0.180 g; 0.366 mmol) in benzene (2.5 mL) was added dropwise. The resulting mixture was stirred at room temperature for 2 h, after which the solvent was removed in vacuo. The obtained solid residue was washed twice with dry hexane to give an analytically pure white solid.

^1^H NMR (250 MHz, CDCl_3_, 298 K): 1.28 (9H, CCH_3_), 1.66 (9H, CCH_3_), 3.50 (2H, NCH_2_), 7.80 (1H, CH=N), 6.6, 7.1, 7.2, 7.6 (1H, Ar-CH).

^13^C{^1^H} NMR (100 MHz, CDCl_3_, 298 K): 21.4, 29.6, 31.8 (C(CH_3_)_3_), 34.3, 34.9 (C(CH_3_)_3_), 42.2 (CH_3_), 50.9, 51.9, 56.9 (CH_2_), 120.8, 122.8 (Ar-C), 123.3, 123.4, 129.3, 130.5, 132.4 (Ar-CH), 135.8, 140.8 (Ar-C), 154.2, 156.7 (Ar-O), 165.0 (CH).

### 3.3. Representative Procedure for the Copolymerization of Epoxides with Anhydrides

A typical copolymerization was carried out as follows. A Schlenk tube was charged sequentially with the designated amounts of the selected anhydride and epoxide, the catalysts, and PPNCl in proper amounts of dry toluene or dry hexane was used. The mixture was analyzedat required temperature and magnetically stirred for required time, and then cooled to room temperature. The mixture was added dropwise to methanol under rapid stirring. The precipitated polymer was recovered by filtration, washed with methanol, and dried at 30 °C overnight in a vacuum oven. The details of the polymerization are reported in Table 1 and Table 2.

NMR peak assignments for polyesters in Table 1 and Table 2.

The ^1^H NMR spectrum (300 MHz, CDCl_3_, 25 °C) of poly(cyclohexene succinate) Appendix A: 4.75 (bs, CH); 2.51 (bs, 2H, CH_2_); 1.94–1.68 (bm, 4H, CH_2_); 1.30 (bm, 4H, CH_2_). ^13^C NMR spectrum (CDCl_3_, 125 MHz) Appendix A: δ 171.8; 73.7; 30.1; 29.4, 28.3.

The ^1^H NMR spectrum (300 MHz, CDCl_3_, 25 °C) of poly(propylene phatalate): δ 7.74–7.63 (bm, 2H); 7.46 (bm, 2H); 5.48 (bm, 1H); 4.44–4.32 (bm, 2H); 1.34 (d, J = 6.6 Hz, 3H). ^13^C NMR spectrum (CDCl_3_, 125 MHz): δ 166.7; 166.8; 132.3; 131.4; 131.8; 131.2; 129.1; 128.9; 69.6; 67.0; 16.4. 

The ^1^H NMR spectrum (300 MHz, CDCl_3_, 25 °C) of poly(limonene phtalate) Appendix A: δ 7.75–7.47 (4H, Ar) 5.53 (1H, OCH); 4.63 (2H, =CH_2_). 2.75, 2.34, 2.23, 1.82–1.53, 1.48, 1.46.

The ^13^C NMR spectrum (100 MHz CDCl_3_) of poly(limonene phtalate) Appendix A: α 165.1–166.6 (2 C). 148.7 (C), 133.1 (CH), 131.3 (2 CH), 128.8 (CH), 125.7 (CH), 109.1 (CH_2_), 82.6 (C), 37.5 (CH), 30.8 (CH_2_), 25.9 (CH_2_), 21.8 (CH_3_), 21.0 (2 C).

The ^1^H NMR spectrum (300 MHz, CDCl_3_, 25 °C) of poly(cyclohexene phtalate): δ 7.61 (2H, Ar); 7.48 (2H, Ar); 5.21 (2H, OCH); 2.43 (2H, CH_2_); 1.83 (2H, CH_2_); 1.53 (4H, CH_2_); 1.43 (2H, CH_2_).

### 3.4. Synthesis of Oly(propylene maleate)-block-polyglycolide

The terpolymerization was carried by charging, sequentially, a Schlenk tube with 2.0 mmol of maleic anhydride (MA), 2.0 mmol of glycolide (GL), an excess of propylene oxide (1 mL), and 0.01 mmol of chromium catalyst **Cr4**. The mixture was then stirred at 50 °C for 22 h, after which it was added dropwise to methanol under rapid stirring. The precipitated polymer was recovered by filtration, washed with methanol, and dried at 30 °C overnight in a vacuum oven.

## 4. Conclusions

In this work, we explored the reactivity and selectivity of salalen chromium and aluminum complexes in the copolymerization of anhydrides, such as succinic anhydride (SA) and phthalic anhydride (PA), with different epoxides, namely cyclohexene oxide (CHO), propylene oxide (PO), and limonene oxide (LO). The polymers obtained showed, in all cases, alternating structures: the percentages of ester junctions were always higher than 99%, indicating that all the investigated catalysts were able to effectively control the selectivity of the reactions. Despite the chiral nature of complexes **Cr2** and **Al2**, no stereoselectivity was observed in the copolymerization of meso epoxides such as CHO.

In the copolymerization of the PO with the PA or MA, the reactions were not regioselective, excluding any possibility of control over the stereochemistry in the polymer. The most active chromium complexes, **Cr2** and **Cr4**, were also tested in the copolymerization of maleic anhydride (MA) with propylene oxide (PO) to produce polypropylene maleate.

The complex **Cr4** was also shown to improve the one-pot terpolymerization of the PO, MA, and glycolide, resulting in the formation of poly(propylene maleate)-*block*-polyglycolide through the one-pot procedure.

This approach makes it possible to simplify the preparation of a diblock copolyester, which is expected to be of value for the preparation of materials with tailored properties for biomedical applications.

## Data Availability

The data presented in this study are available on request from the corresponding authors (DP and MM).

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
