# Peer review of "Salen-like Chromium and Aluminum Complexes as Catalysts in the Copolymerization of Epoxides with Cyclic Anhydrides for the Synthesis of Polyesters"

_ijms, 2023, doi:10.3390/ijms241210052_

Round 1

Reviewer 1 Report

This Pappalardo and Mazzeo et al paper describes an interesting study about copolymerization behaviour of a various Cr and Al salalen catalyst under different conditions with different monomers for the preparation of materials properties for biomedical applications. The work is carried out very nicely and the compounds are very well characterized with different techniques although its paramagnetic nature in chromium compounds. The well-known experience of these researchers in the field of polymerization has been demonstrated in many articles and recently in a review (DOI 10.1002/ejic.202200644

The characterization of the catalyst have been done and copolymerization of anhydrides such with different epoxides have been deeply study from a regio-or steroeselecivity point of view.

I recommended this manuscript for publication with minor revisions related with typographical mistakes.

·       Scheme 1 could be better named as Figure 1.

·       In this Scheme 1. Mt is not a suitable abbreviation because it is the name of a chemical compound called meitnerium, perhaps it would be better to put M.

·       Line 180; scheme 4 must be written in capital letters, Scheme 4.

·       Line 220: a space is required between the number and the grades, 130 oC.

·       Line 246: AlMe3, but it could be AlMe3 with subindex

·       Line 339; polypropylene instead of poly propylene.

·       Line 382; I believe that a pre-prosecution is needed. “For the synthesis of a…”

·       Line 397 Merck instead of Merk

Thank you so much

Dra. Vanessa Tabernero Magro

This Pappalardo and Mazzeo et al paper describes an interesting study about copolymerization behaviour of a various Cr and Al salalen catalyst under different conditions with different monomers for the preparation of materials properties for biomedical applications. The work is carried out very nicely and the compounds are very well characterized with different techniques although its paramagnetic nature in chromium compounds. The well-known experience of these researchers in the field of polymerization has been demonstrated in many articles and recently in a review (DOI 10.1002/ejic.202200644

The characterization of the catalyst have been done and copolymerization of anhydrides such with different epoxides have been deeply study from a regio-or steroeselecivity point of view.

I recommended this manuscript for publication with minor revisions related with typographical mistakes.

·       Scheme 1 could be better named as Figure 1.

·       In this Scheme 1. Mt is not a suitable abbreviation because it is the name of a chemical compound called meitnerium, perhaps it would be better to put M.

·       Line 180; scheme 4 must be written in capital letters, Scheme 4.

·       Line 220: a space is required between the number and the grades, 130 oC.

·       Line 246: AlMe3, but it could be AlMe3 with subindex

·       Line 339; polypropylene instead of poly propylene.

·       Line 382; I believe that a pre-prosecution is needed. “For the synthesis of a…”

·       Line 397 Merck instead of Merk

Thank you so much

Dra. Vanessa Tabernero Magro

Author Response

ANSWER: We thank the reviewer for the time and concern in reviewing the manuscript. All the comments have been considered and corrections applied.

  • Scheme 1 could be better named as Figure 1.

ANSWER: Thank you for the comment. However, we prefer to use the word “Scheme” for drawing of the structures of the compounds and for the reaction schemes, that were made by using the ChemDrawn program.

  • In this Scheme 1. Mt is not a suitable abbreviation because it is the name of a chemical compound called meitnerium, perhaps it would be better to put M.

ANSWER: Thank you for the comment. The abbreviation was changed accordingly.

  • Line 180; scheme 4 must be written in capital letters, Scheme 4.
  • Line 220: a space is required between the number and the grades, 130 oC.
  • Line 246: AlMe3, but it could be AlMe3with subindex
  • Line 339; polypropylene instead of poly propylene.
  • Line 382; I believe that a pre-prosecution is needed. “For the synthesis ofa…”
  • Line 397 Merck instead of Merk

ANSWER: Thank you for the comment. All the corrections have been carried out accordingly to the reviewer suggestions.

Reviewer 2 Report

The paper is interesting, because it opens the route to new interesting biomedical polymers.

However, the presentation of the results should be improved.

1) The mechanism of polymerization (coordination-insertion or some else) should be discussed and proved somehow by the data or literature. How the complex structure was designed to allow involvement of two monomers with different ability to polymerize?

2) As authors deal with copolymerization, the initial comonomers ratios and resulting ratios of monomer units within the polymer should be given. 

3) the NMR data should be given for all synthesized copolymers

4) Methods part should be substantially improved. The materials and equipment details should be given. Also the synthetic procedures allowing the production of the polymers should be described. Methods of polymers characterization should be also described

5) Authors should discuss better the perspectives for increasing the molecular weight of the polymers, because it is currently too low for preparation of any biomedical materials

Author Response

The paper is interesting, because it opens the route to new interesting biomedical polymers. However, the presentation of the results should be improved.

ANSWER: We thank the reviewer for the time and concern in reviewing the manuscript. All the comments have been considered and corrections applied.

  • The mechanism of polymerization (coordination-insertion or some else) should be discussed and proved somehow by the data or literature. How the complex structure was designed to allow involvement of two monomers with different ability to polymerize?

ANSWER:  The author thanks the reviewer for this suggestion. Scheme 5 and an explanation of the polymerization mechanism was added in the paper. 

  • As authors deal with copolymerization, the initial comonomers ratios and resulting ratios of monomer units within the polymer should be given. 

ANSWER: The initial comonomers ratios has been reported in the Table 1 and Table 2. Since the copolymers described in tables 1 and 2 are formed by alternating copolymerization of epoxide and anhydride, the final composition is 1:1. Moreover, the structure of the copolymer is described by other important parameters, the conversion of the anhydride (Conv. %), and the percentage of the ester linkage in the polymer (ester %). This information has been added in the caption of Table 1, for the sake of clarity.  

  • the NMR data should be given for all synthesized copolymers

ANSWER:  The NMR data were added for all polymers.

4) Methods part should be substantially improved. The materials and equipment details should be given. Also the synthetic procedures allowing the production of the polymers should be described. Methods of polymers characterization should be also described

ANSWER: Thank you for the comment. This part has been improved as suggested. Missed information have been added.

5) Authors should discuss better the perspectives for increasing the molecular weight of the polymers, because it is currently too low for preparation of any biomedical materials

ANSWER: Although high molecular weight polymers are generally preferred, low molecular weight (oligomeric) PPF is largely used as 3D printable material by stereolithographic processes. For this application viscosities less than ca. 2 Pa s are needed to ensure the printability of structures. PPF exhibits a solid-like behaviour when its molecular mass is greater than 4000 Da and the solubility decreases with increasing chain length. Consequently, only low molecular mass (<3000 Da) oligomers are suitable for these techniques. These comments were added in the paper.

Reviewer 3 Report

In the submitted manuscript by Mazzeo and co-workers, chromium and aluminum complexes bearing salalen ligands are tested as catalysts in the ring-opening copolymerization (ROCOP) of anhydrides and epoxides. Firstly, Catalysts Cr1, Cr2 and Al2 are found to catalyze ROCOP between anhydrides (succinic anhydride (SA) or phthalic anhydride (PA)) and epoxides (cyclohexene oxide (CHO), propylene oxide (PO) or limonene oxide (LO)), showing highly alternating structures under selected conditions. The polymerization results also show that catalyst Cr2 exhibits higher activity than Al2. In addition, ROCOP of maleic anhydride (MA) and PO is compared between Cr2 and a commercial complex Cr4. Finally, terpolymerization of MA, PO and glycolide (GL) catalyzed by Cr4 is realized by switch catalysis between ROCOP and ring-opening polymerization of GL. In all of the above cases, low molecular polymers are obtained and characterized by NMR, GPC and MALDI-TOF. Overall, the authors have demonstrated the catalysis behavior of salalen-Cr and salalen-Al complexes in ROCOP of anhydrides and epoxides. Although these complexes do not show no significant advantages compared to previously reported catalysts, they do enrich the catalyst library in the ROCOP of anhydrides and epoxides. This work could be suitable for Polymers, after the following issues are addressed.

Major points:

·       I do not see the Supplementary Material. Please double-check if it has been submitted together with the main manuscript. I have also informed the editor about this issue.

·       In the introduction, a little more background about salalen-metal complex in polymerization would be helpful to show difference between salalen ligands and the common salen ligands.

·       Table 1 – How does Al2 compare with Cr2 in the copolymerization between SA and CHO, PO or LO? These results could further showcase the activity comparison between Al2 and Cr2.

·       Table 1, entries 1 and 2 – Can PPNCl be used in these experiments instead of DMAP, and would this change suppress the formation of ether linkage? Experiments using PPNCl as co-catalyst are recommended.

·       Can the salalen-based Cr1, Cr2 or Al2 be used in the switch catalysis to catalyze terpolymerization of MA, PO and GL?

·       What is the molecular weight after the isomerization of PPM? GPC of the produced PPF should be included and compared with the GPC of PPM.

Minor points

·       Since the effect of water is crucial in ROCOP, are all the epoxides and anhydrides purified or dried before polymerization?

·       Line 108 – “non” should be “not”

·       Line 132-134 – A reference paper on the previously reported synthesis would be helpful.

·       Scheme 1 ­– Why do Cr2 and Al2 have asymmetric ligand with one aromatic ring containing chlorine substitution?

·       Scheme 1 – The structure of Al2 is incorrect. According to scheme 3, Al2 contains a methyl group instead of a chlorine.

·       Line 146-147 – Is  “Cr1” actually “Cr2” in this sentence?

·       Line 193 – I think “the makes up” should be “it makes up”.

·       Table 1 – Can TOF values also be listed in table 1?

·       Line 234 & 260 – I think “perfectly” is too strong. Consider toning down the language.

·       Line 256 – The note 41 could be removed, since it is repeated in the next sentence.

·       Discussion of table 2, entry 5 should be added in the main text.

·       Figure 4 – While the MALDI-TOF spectrum suggests OH-terminated structure, I wonder why there is no Cl-terminated structure since PPNCl is used as co-catalyst in the polymerization.

·       Line 340 – “CHCl3” should be “CDCl3”.

·       Line 361 – I think “either” should be “both”.

·       In the terpolymerization of MA, PO and GL, what is the incorporation ratio of two polyester blocks based on the NMR spectrum in figure 7?

·       Line 412 – Is “salen” actually “salalen” here?

The manuscript is well-written, easy to understand. The data and resutls are clearly presented.

Author Response

The authors thank the reviewer for her/his comments and suggestions that helped us to improve the quality of the manuscript.

Major points:

  • I do not see the Supplementary Material. Please double-check if it has been submitted together with the main manuscript. I have also informed the editor about this issue.

ANSWER: The authors thank the reviewer for noting this problem, we will try to solve it uploading again the file of  Supplementary Material.

  • In the introduction, a little more background about salalen-metal complex in polymerization would be helpful to show difference between salalen ligands and the common salen ligands.

ANSWER: Thank you for the comment. Some sentences and the related references about  comparison between salen and salalen complexes in this catalysis have been added in the introduction.

  • Table 1 – How does Al2 compare with Cr2 in the copolymerization between SA and CHO, PO or LO? These results could further showcase the activity comparison between Al2 and Cr2.

ANSWER:  Following the reviewer’s suggestion a polymerization experiment of SA and CHO with Al2 was performed.  The data were added in Table 1 as entry 6.  

  • Table 1, entries 1 and 2 – Can PPNCl be used in these experiments instead of DMAP, and would this change suppress the formation of ether linkage? Experiments using PPNCl as co-catalyst are recommended.

ANSWER: Thank you for the comment. A request by the reviewer an additional polymerization experiment ( entry 2, Table 1) was performed with complex Cr1 and PPNCl under the same reaction conditions of entry 1. In this case the activity was drastically reduced while the selectivity improved. These data were added in Table 1 as run 2.

  • Can the salalen-based Cr1, Cr2 or Al2 be used in the switch catalysis to catalyze terpolymerization of MA, PO and GL?

ANSWER: We tried to perform the terpolymerization of MA, PO and GL with Al2 catalyst without obtained results because the aluminum complex is very sensitive to the protic impurities contained into the glycolide. The ability of complex Al2 to promote a switch catalysis was confirmed by terpolymerization reaction of MA, PO and lactide. These data were not included because in this paper because they are object of a different study that we are performing. 

  • What is the molecular weight after the isomerization of PPM? GPC of the produced PPF should be included and compared with the GPC of PPM.

ANSWER: We verified that the MM of polypropylene maleate does not change after isomerization. The same result was observed by other authors and by us see ref 23 and ref 26. A comment and the related references were added in the paper.

Minor points

  • Since the effect of water is crucial in ROCOP, are all the epoxides and anhydrides purified or dried before polymerization?

ANSWER: All anhydrides have been purified by extraction and crystallization from toluene. The epoxides were distilled over CaH2.  Because of their high hydrophilicity of the anhydrides, the procedure reported in the literature for their purification  is very long: extraction with toluene, twice crystallizations from CH2Cl2 and subsequent sublimation. See for example SI of the paper  J. Am. Chem. Soc. 2015, 137, 38, 12179–12182.

  • Line 108 – “non” should be “not”

ANSWER: thank you for noting this mistake. It was corrected.

  • Line 132-134 – A reference paper on the previously reported synthesis would be helpful.

ANSWER: The reference was added.

  • Scheme 1 ­– Why do Cr2 and Al2 have asymmetric ligand with one aromatic ring containing chlorine substitution?

ANSWER:  in a previous study we demonstrated that the presence of halogen atoms on the ligand skeleton has beneficial effect on the catalytic activity in the ROP of lactide (see ref 40) thus we decided to use these ligands.

  • Scheme 1 – The structure of Al2 is incorrect. According to scheme 3, Al2 contains a methyl group instead of a chlorine.

ANSWER: Thank you for noting this mistake, the scheme was modified.

  • Line 146-147 – Is  “Cr1” actually “Cr2” in this sentence?

ANSWER: Thank you for noting this mistake, it was corrected.

  • Line 193 – I think “the makes up” should be “it makes up”.

ANSWER: The mistatke was corrected

  • Table 1 – Can TOF values also be listed in table 1?

ANSWER: The TOF values were added in Table 1. 

  • Line 234 & 260 – I think “perfectly” is too strong. Consider toning down the language.

ANSWER: Following the reviewer’ suggestion, the adjective “perfectly” was removed. 

  • Line 256 – The note 41 could be removed, since it is repeated in the next sentence.

ANSWER: The note was removed. 

  • Discussion of table 2, entry 5 should be added in the main text.
  • Figure 4 – While the MALDI-TOF spectrum suggests OH-terminated structure, I wonder why there is no Cl-terminated structure since PPNCl is used as co-catalyst in the polymerization.

ANSWER: Signals of lower intensities corresponding to Cl-terminated chains were detected, see for example the signal at 1791.70  m/z in figure 4.  

  • Line 340 – “CHCl3” should be “CDCl3”.

ANSWER: This mistake was corrected

  • Line 361 – I think “either” should be “both”.

ANSWER: This mistake was corrected

  • In the terpolymerization of MA, PO and GL, what is the incorporation ratio of two polyester blocks based on the NMR spectrum in figure 7?

ANSWER: The two blocks have the same length, coherently with the feed ratio. This information was added in the paper.

  • Line 412 – Is “salen” actually “salalen” here?

ANSWER: This mistake was corrected

Round 2

Reviewer 2 Report

Authors provided manuscript with substantial impovements. However, manuscript is still inaccurately designed and have many unclear places.

When preparing the response to the reviewer authors should point out the lines in the manuscript where the corrections are contained. Otherwise it is hard to analyse the revision.

1. NMR Figures. The protons designations should be unified and presented according to the common rules. Small letters and starting from upfield protons signals to downfield. The solvent used for spectra registration should be given in the caption.

2. Scheme 6 should contain the conditions of reaction

3. Table 2 Why authors used only equimolar ratios? Column 3 should be excluded, because there is no change. Authors provided only composition of the reaction mixture, however the composition of polymer should be also provided.

4. Table 2. Why molecular weight for the polymer with highest conversion, which was obteined without solvent, was not determined?

The quality of English should be improved.

Author Response

Authors provided manuscript with substantial improvements. However, manuscript is still inaccurately designed and have many unclear places.

When preparing the response to the reviewer authors should point out the lines in the manuscript where the corrections are contained. Otherwise it is hard to analyse the revision.

The author thank the reviewer for these suggestions. 

  1. NMR Figures. The protons designations should be unified and presented according to the common rules. Small letters and starting from upfield protons signals to downfield. The solvent used for spectra registration should be given in the caption.

- ANSWER: The NMR figures were modified according the reviewer’s indications.

  1. Scheme 6 should contain the conditions of reaction :

- ANSWER: We thank the reviewer for the suggestion, and we agree that a reaction scheme should contain the conditions. However, in this case, the aim was to present a general reaction, while the specific conditions are detailed in table 2. As a matter of fact, the polymerization conditions were different in all the polymerization runs, thus we have decided to refer to the table 2. For the sake of clarity, a sentence was added in the caption of Scheme 6, please see p. 8, lines 308-309.

  1. Table 2 Why authors used only equimolar ratios? Column 3 should be excluded, because there is no change. Authors provided only composition of the reaction mixture, however the composition of polymer should be also provided.

- ANSWER: We thank the reviewer for the observation, and we have modified the table 2 accordingly to the suggestion.

The composition of the polymer is dictated by the chemical structure; the product is always a perfect propylene maleate due to the alternated enchainment of PO and MA. No ether sequences are observed, while the percentage of ester bonds is more than 99 %. For the sake of clarity, a further sentence has been added in the text. See p. 8, lines 313-315

  1. Table 2. Why molecular weight for the polymer with highest conversion, which was obtained without solvent, was not determined?

ANSWER: The molecular masses were performed, and data were added.

Reviewer 3 Report

I'm glad to see that the authors have addressed my concerns and provided corresponding explanations in their latest manuscript revision. A supplementary material has also been updated. Based on these revisions, I believe that the manuscript is now suitable for publication, after the following minor issues are addressed.

·       Line 153 – Should “Cr1” also be changed to “Cr2”?

·       Line 157 & 162 – Use “Figure S2”, “Figure S3” instead of “Fig. S2”, “Fig. S3”

·       Line 272 – I think “entries 5-8” needs to be updated (entries 7-10?).

·       Line 273 – I think “Figure S2” needs to be updated (Figure S5 or S6?).

·       Line 365 – Figure S3 should be Figure S8.

·       Supplementary Material – Make sure the sample labels for Figure S5-S7 are correctly showed. Currently both Figure S6 and S7 are the 13C NMR spectra of CHO/PA copolymer obtained in entry 5 in Table 1.

Author Response

We thank the reviewer for the time and concern in reviewing the manuscript. All the comments have been considered and corrections applied.

The modifications are highlithed in purple in the text.